# Mid-Infrared (MIR) Complex Refractive Index Spectra of Polycrystalline Copper-Nitride Films by IR-VASE Ellipsometry and Their FIB-SEM Porosity

Emilio Márquez [1,*], Eduardo Blanco [1], José M. Mánuel [1], Manuel Ballester [2], Marcos García-Gurrea [1], María I. Rodríguez-Tapiador [3], Susana M. Fernández [3], Florian Willomitzer [4] and Aggelos K. Katsaggelos [5]

[1] Department of Condensed-Matter Physics, Faculty of Science, University of Cadiz, 11510 Puerto Real, Spain; eduardo.blanco@uca.es (E.B.); jose.manuel@uca.es (J.M.M.); marcos.garciagu@alum.uca.es (M.G.-G.)
[2] Department of Computer Sciences, Northwestern University, 633 Clark St., Evanston, IL 60208, USA; manuelballestermatito2021@u.northwestern.edu
[3] Renewable Energy Department, CIEMAT, Avenida Complutense 40, 28040 Madrid, Spain; mariaisabel.rodriguez@ciemat.es (M.I.R.-T.); susanamaria.fernandez@ciemat.es (S.M.F.)
[4] Wyant College of Optical Sciences, University of Arizona, Tucson, AZ 85721, USA; fwillomitzer@arizona.edu
[5] Department of Electrical and Computer Engineering, Northwestern University, Evanston, IL 60208, USA; a-katsaggelos@northwestern.edu
* Correspondence: emilio.marquez@uca.es

**Abstract:** Copper-nitride ($Cu_3N$) semiconductor material is attracting much attention as a potential, next-generation thin-film solar light absorber in solar cells. In this communication, polycrystalline covalent $Cu_3N$ thin films were prepared using reactive-RF-magnetron-sputtering deposition, at room temperature, onto glass and silicon substrates. The very-broadband optical properties of the $Cu_3N$ thin film layers were studied by UV-MIR (0.2–40 μm) ellipsometry and optical transmission, to be able to achieve the goal of a low-cost absorber material to replace the conventional silicon. The reactive-RF-sputtered $Cu_3N$ films were also investigated by focused ion beam scanning electron microscopy and both FTIR and Raman spectroscopies. The less dense layer was found to have a value of the static refractive index of 2.304, and the denser film had a value of 2.496. The iso-absorption gap, $E_{04}$, varied between approximately 1.3 and 1.8 eV and could be considered suitable as a solar light absorber.

**Keywords:** optical properties; thin films; electron microscopy; spectroscopic ellipsometry; solar energy

## 1. Introduction

There is a need to innovate in eco-friendly, advanced materials to provide the answer to the social demand for sustainable energy [1–3]. Determination and understanding of the optical properties of covalent polycrystalline copper-nitride ($Cu_3N$) thin films, such as refractive index and extinction coefficient and bandgap energy, are important to carry out a photovoltaic-cell design, in which the $Cu_3N$ material acts as solar light absorber [4,5]. It would open the door to the next, flexible third generation of photovoltaic technologies that could benefit from this material. The practical application of copper-nitride layers mainly depends upon the size of its optical bandgap. This nitride is a non-toxic choice to consider as a possible alternative for tellurium-based materials.

$Cu_3N$ thin films show highly promising potential in the field of solar energy. Their cubic crystal structure with a lattice parameter of $a = 3.815$ Å and a density of 5.9 g/cm$^3$, along with a chemical composition of Cu-N-Cu and bond angles of approximately 180°, offers properties that can be adjusted by properly controlling technical parameters during the preparation. This allows for the optimization of the optical band gap of the films, especially in the visible light range [6], and the attainment of maximum photovoltaic voltages [7]. Depositing $Cu_3N$ (100) thin films, both *p*-type and *n*-type, on substrates such

as SrTiO$_3$ (100) by using various technologies facilitates the introduction of *p*-type and *n*-type impurities into Cu defects. This approach not only contributes to the formation of photovoltaic materials with a satisfactory band gap, but also provides the opportunity to adjust and enhance the optical and electrical properties through precise control of the Cu/N chemical composition during thin film deposition. In this context, the bipolar doping of Cu$_3$N thin layers emerges as a promising strategy to notably increase the efficiency of solar energy conversion [8,9].

Paradoxically, despite the great expectations that the *metastable* covalent indirect-gap Cu$_3$N semiconductor is awakening because of its optical and energy-storage properties, it is *not* yet employed in a specific solar cell. The development of a low-cost Cu$_3$N semiconductor, free of critical materials, and prepared with easy growth techniques for industrial scaling, such as reactive-RF-magnetron-sputtering deposition, is nowadays considered a hot topic in emerging-technology photovoltaics. In other words, more research is still needed because there is yet relatively scarce information about the potential use of Cu$_3$N as an element of a solar cell.

The present communication reports the feasible and successful preparation of the Cu$_3$N binary compound, with an *anti*-ReO$_3$ cubic crystal structure, importantly, at room temperature, onto glass and silicon substrates, and using two different gaseous environments. Utilizing the reactive magnetron sputtering technique, copper (metal target) and nitrogen (the reactive gas) might be combined with argon as an inert working gas. This common method allows for the formation of thin films with varied chemical compositions, facilitating the creation of diverse compounds from a single metallic target [10]. In this work, we employed both (*i*) an Ar-free environment based only upon nitrogen (N$_2$), and (*ii*) a mixture of N$_2$ and Ar, for the fabrication of copper-nitride thin films, demonstrating the application of the reactive-magnetron-sputtering theory in practice [11–15].

Focused ion beam scanning electron microscopy (FIB-SEM) was employed to determine the surface morphology of the Cu$_3$N thin-layer samples under study. There is a clear interest in the analysis of porosity in the present Cu$_3$N films by this FIB-SEM technique. It has been considered for the production of Cu$_3$N layers, to take advantage of the existing porosity of the copper-nitride layers grown at oblique angle.

We then calculated the complex refractive index, $\tilde{n} = n + ik$ ($n$ is the real refractive index and $k$ is the extinction coefficient), of the copper-nitride layers employing UV-MIR spectroscopic ellipsometry [16,17], for the first time, to the best of our knowledge. It should be emphasized that in this study, as the main novelty spectro-ellipsometric measurements in the infrared spectral range were additionally performed at up to 40 µm, in which molecular vibrations and both free-charge-carrier and *phonon* (lattice) absorption are probed, thus providing very valuable and diverse information about the material. It has also to be stressed that the UV-visible spectral range is just sensitive to the electronic states and excitons. Besides information about the layer thickness, modeling of the IR ellipsometric spectra therefore provides useful information on the chemical, structural and infrared properties of the Cu$_3$N thin films.

## 2. Experimental Procedure

The Cu$_3$N thin films were manufactured at room temperature through a monochamber sputtering process using the commercial MVSystems LLC (Golden, CO, USA), connected to a 13.56 MHz, 600 W radio-frequency power source from Coaxial Power Systems Ltd. (Eastbourne, UK). The Cu target used in the process had a diameter of 3 inches and a purity level of 99.99%. It was sourced from Lesker Company (St. Leonards-on-Sea, East Sussex, UK). The sputtering chamber was evacuated to a base pressure of approximately $10^{-5}$ Pa, and the sputtering process occurred in an environment with partial pressures of N$_2$ at 0.8 and 1.0, defined as R = [N$_2$]/([N$_2$] + [Ar]). The total working pressure, set at 5.0 Pa, was controlled by a 'butterfly' valve to ensure precision in the process. Figure 1a shows a photograph of the MVSystems LLC equipment, where the produced close-to-reddish-colored RF plasma is shown, while Figure 1b displays the schematic image of the RFMS

process. In this procedure, the gases used were nitrogen ($N_2$) with a purity of 99.999% and argon (Ar) with a purity of 99.99995%. The flow rate of both gases was accurately controlled at 20 sccm and 10 sccm, respectively, using a mass flow controller (MFC) from MKS Instruments, (Andover, MA, USA). More details regarding the deposition conditions are listed in Table 1.

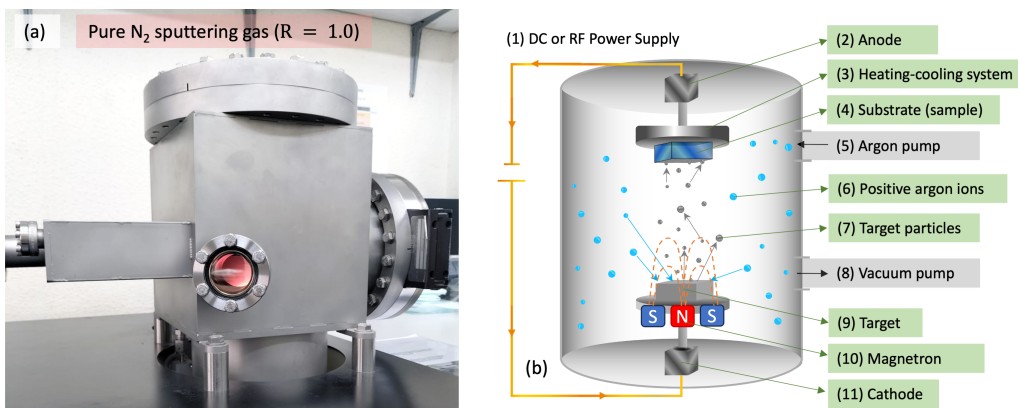

**Figure 1.** (**a**) Reactive-RF-magnetron-sputtering chamber, where the RF plasma generated by the emissions from the excited atoms and ions is seen. (**b**) Schematic image of the sputtering process.

**Table 1.** Deposition conditions for $Cu_3N$ thin films and their very broadband optical properties.

| Sample ID | $N_2$ Flux (sccm) | Ar Flux (sccm) | Partial $N_2$ Pressure | Deposition Time (min) | Total Oscillators (TL + Gaussian) | Sample Thickness (nm) | Ellipsometry Sample Roughness (nm) | Microscopy Sample Roughness (nm) | Urbach Energy (meV) |
|---|---|---|---|---|---|---|---|---|---|
| #1360 | 20 | 10 | 0.8 | 60 | **12** (1 + 11) | **430** | **42** | **19** | **96** |
| #1460 | 20 | 0 | 1.0 | 60 | **9** (1 + 8) | **333** | **22** | **7** | **176** |
| #1490 | 20 | 0 | 1.0 | 90 | **7** (1 + 6) | **610** | **46** | **20** | **242** |

Focused ion beam scanning electron microscopy was employed to study the topography of $Cu_3N$ layers. The focused-ion-beam technique was used to obtain transversal trenches and remove material from the surface, for measuring pore sizes, using the software 'ImageJ'. Further details are found elsewhere [5].

UV-Visible-NIR spectroscopic ellipsometry (SE) measurements were used to acquire the ellipsometric angles $\Psi$ and $\Delta$, on a Woollam vertical variable-angle-of-incidence rotating-analyzer ellipsometer. Data were obtained at three angles of $50°$, $60°$, and $70°$, respectively. Novel infrared spectroscopic ellipsometry (IRSE) measurements were also carried out on a Woollam IR-VASE Mark II ellipsometer, integrating a Fourier-transform infrared interferometer source. The experimental SE and IRSE data were modeled using the WVASE software package, version 3.942. FTIR measurements were performed using a Perkin-Elmer 100 FTIR spectrometer. Raman measurements were carried out employing a dispersive spectrometer Horiba-Jobin-Yvon LabRam HR 800. Normal-incidence optical transmission was also measured using a double-beam spectrophotometer (Lambda 1050 ultraviolet-visible-near infrared spectrometer, Perkin Elmer).

## 3. Results and Discussion

### 3.1. FIB-SEM Microscopy Study

The $Cu_3N$ thin films exhibited a columnar formation, as shown in Figure 2a (the maximum and the minimum values of the film thickness are indicated in the SEM micrograph). It is observed that the copper-nitride microstructure through about the first 100 nm (samples #1460 and #1490), or around the first 200 nm (sample #1360), from the glass surface, is undoubtedly compacted, while voided spaces between the $Cu_3N$-layer columns are clear in the rest of the layer thickness. This is the columnar structure 'zone 2' of the Thornton structural zone model [18]; it consists of columnar and compact grains, with high density and smooth surfaces.

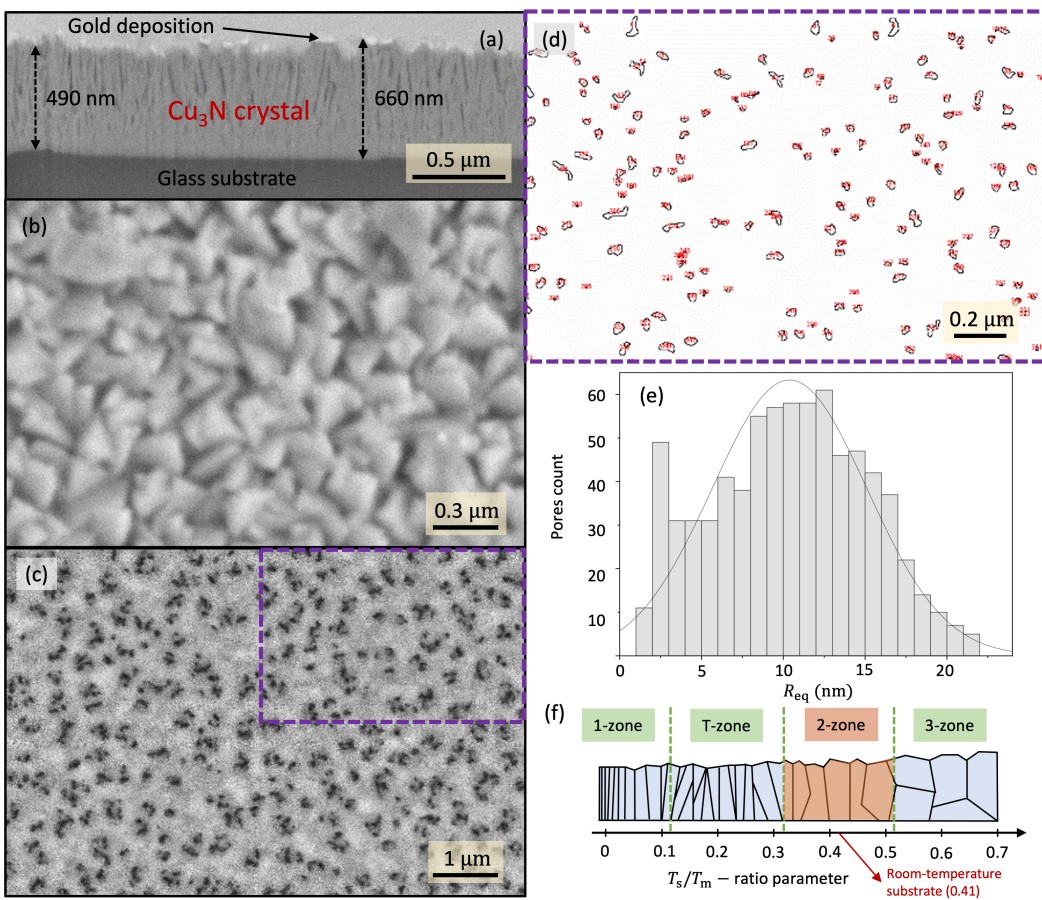

**Figure 2.** SEM micrographs of (**a**) cross-section and (**b**) planar views, both of sample #1490. (**c**) SEM image revealing the internal porosity of sample #1360, and (**d**) a pore size map obtained using 'ImageJ' software, with (**e**) its associated histogram for the *pore equivalent radii*. (**f**) The well-known Thornton structural zone model is presented for comparison.

It is verified that the $T_s/T_m$ ratio (a parameter establishing the corresponding 'structural zone') must obey the condition, $0.3 < T_s/T_m < 0.5$, in this particular zone 2, where $T_s$ is the substrate temperature during deposition, and $T_m$ is the coating-material melting point, which is well known to be above 740 K, the decomposition temperature of $Cu_3N$. Hence, it is clear that the columnar structure of $Cu_3N$ results from being grown at room temperature and low working-gas pressure. The value of the $T_s/T_m$ ratio in our RF-sputtering depositions is found to be 0.41, certainly a value corresponding to the zone 2 category. Figure 2b displays an SEM micrograph of the surface of sample #1460. $Cu_3N$ pillars met at its surface, giving place to a conglomerated structure with many sealed 'closed' pores.

The FIB-SEM porosity of sample #1360 (see Figure 3a, where a representative photo of a close-to-crimson-colored $Cu_3N$ layer is also displayed) was quantified using images such as the one in Figure 2c. The pore map in Figure 2d, on the other hand, was obtained using the software 'ImageJ' from the region indicated by a dashed frame in Figure 2c. This software is commonly utilized to measure particle sizes from images. It has similarly and successfully been used in this work to obtain the *equivalent pore radii*. Figure 2d shows the pore perimeter (black lines), and the numbering (in red) that 'ImageJ' uses to identify each particular pore. Pore areas, $A_{pore}$'s, were automatically measured from the map, and the corresponding values of the equivalent pore radius, $R_{eq}$, were obtained using the expression: $R_{eq} = \sqrt{A_{pore}/\pi}$. Figure 2d displays the corresponding histogram for the values calculated from the SEM image in Figure 2c; the mean equivalent radius in this particular sample was found to be $10.4 \pm 4.7$ nm. Hence, the present $Cu_3N$ films were found to have a low porosity, which is desirable for photovoltaic applications.

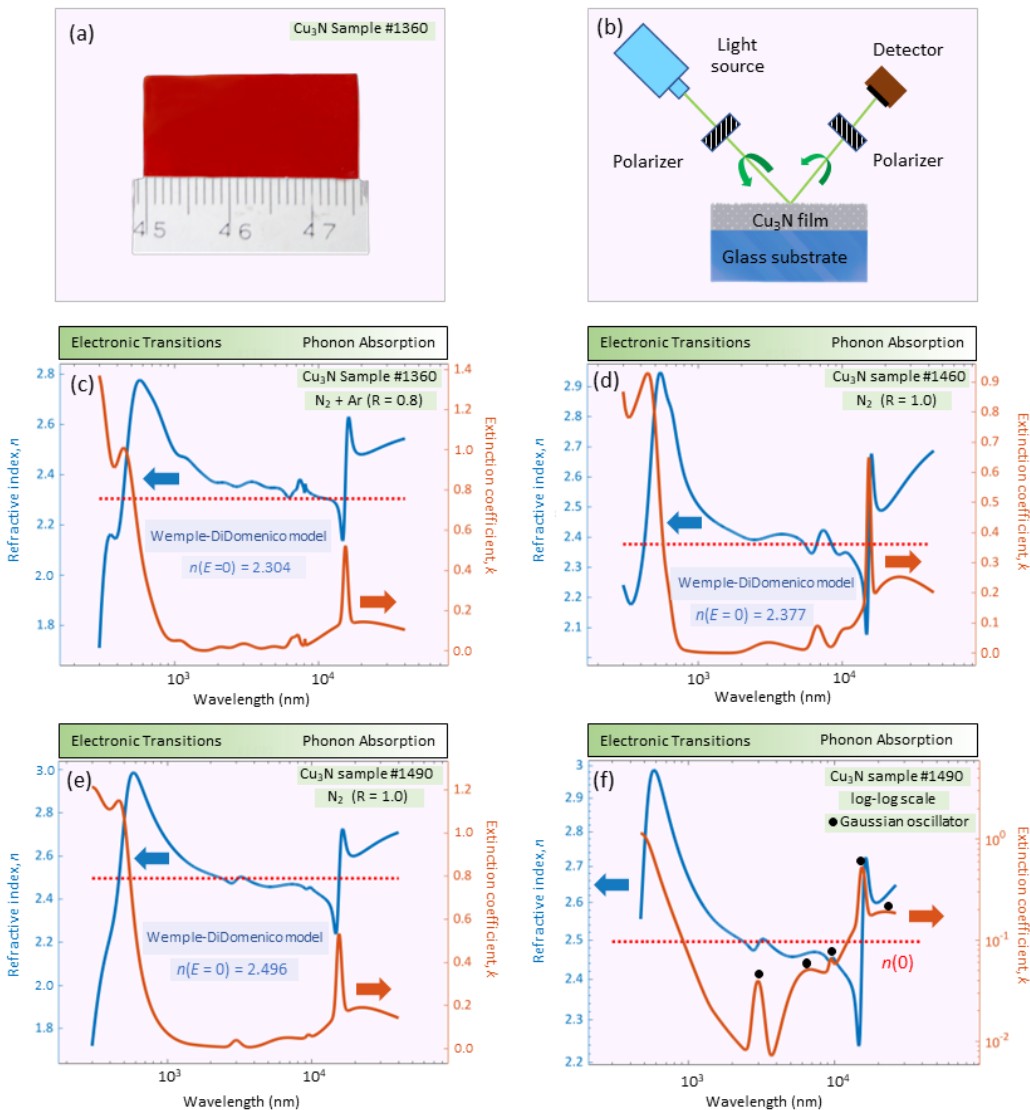

**Figure 3.** (**a**) A representative sample photo, and (**b**) a schematic ellipsometry set-up. Optical functions *n* and *k* of samples (**c**) #1360, (**d**) #1460, (**e**) #1490. (**f**) The log-log scale of sample #1490.

It is emphasized that the porous nature of $Cu_3N$-layer inner microstructure is not a consequence of a non-reliable deposition technique, but the result of the growth mode of the thin-film material. It follows from zone 2 of the Thornton model, which indicates a columnar growth. Nevertheless, as it can be observed in the SEM images from the $Cu_3N$-material surface (see Figure 2), the porosity of the layer is highly reduced when its surface is formed. In this way, the functionality of this material as an active layer for solar cells is not spoiled at all by the porosity observed between the substrate/$Cu_3N$ interface and the $Cu_3N$-film surface.

### 3.2. UV-MIR Ellipsometric Analysis

Optical and infrared ellipsometric data (see Figure 3b) were fit over the ample range of 200–40,000 nm (0.031–6.2 eV), simultaneously with normal-incidence optical transmission data (200–2500 nm, or 0.5–6.2 eV). This is another novelty of this work compared to [5], which was not mentioned above. The best-fit ellipsometric model required, in the case of sample #1490, the introduction of a 46 nm thick surface-roughness layer (Table 1).

Figure 3 shows the best-fit very-broadband optical constants, *n* and *k*, for the $Cu_3N$ samples. Table 1 gives information about the necessary number of oscillators in order to

predict the dielectric functions contained in the so-called WVASE *GenOsc* layer for all Cu$_3$N samples. The excellent comparison between the surface roughness determined by both spectro-ellipsometry and atomic force microscopy is also shown in Table 1. In addition, two major spectral features are seen in the optical constants of Cu$_3$N: A UV-Visible absorption edge with a peak at approximately 2.47 eV, and a second sharp resonant absorption in the infrared near a wavelength of 15,480 nm, both peaks for the particular case of #1490 layer. The specific UV-Visible absorption edge, also with a clear peak, was modeled by combining a Gaussian and a Tauc-Lorentz oscillators [19]. Seven Gaussian oscillators were also added to fit the two ellipsometric angles, Ψ and Δ, in the rest of the spectral range under study, for the #1490 sample (see Table 1). The extremely sharp resonant absorption peak at a wavenumber of 646 cm$^{-1}$ (Figure 4a), on the other hand, determined by ellipsometry, and reported for the first time, suggests that this sample is of polycrystalline nature. It must be indicated that, although not shown in this communication, we have previously performed X-ray diffraction measurements on all the three samples studied in the present work, demonstrating the polycrystallinity of the as-deposited RFMS Cu$_3$N films [5], as we have also concluded from the *k*-spectra displayed in Figure 3. This peak at 646 cm$^{-1}$ was modeled using a Gaussian oscillator, though it could also have been used instead of a Lorentz oscillator.

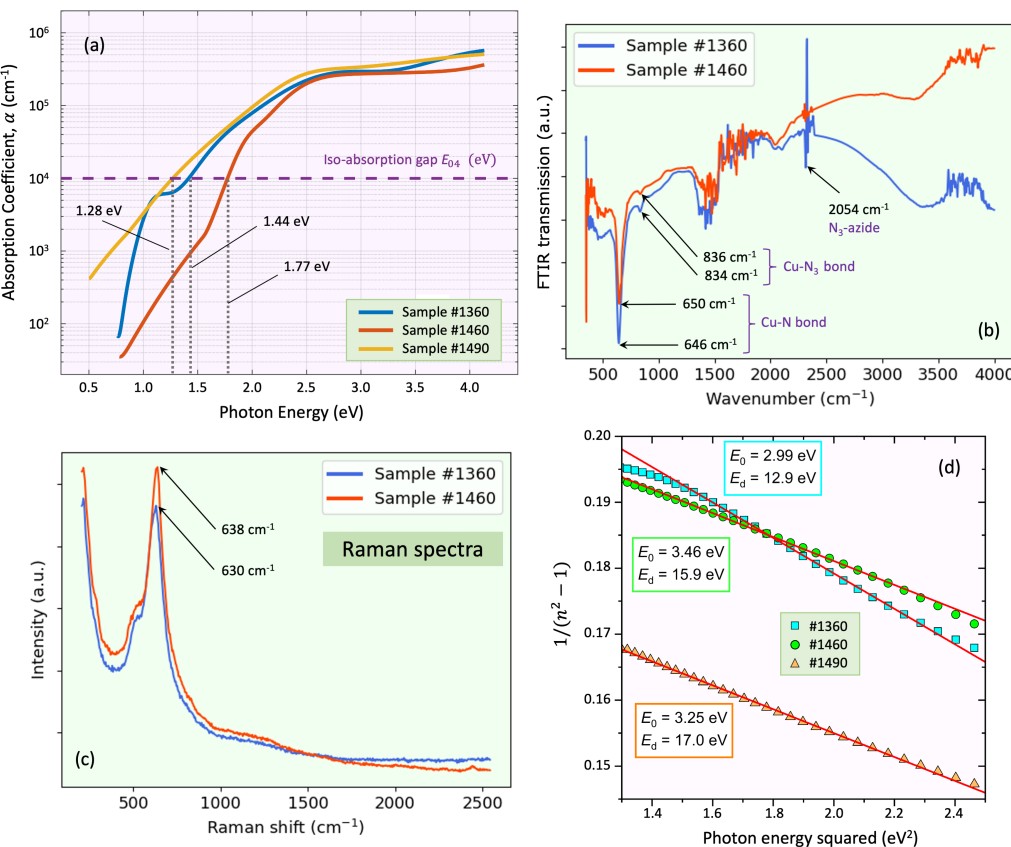

**Figure 4.** (**a**) Optical-absorption edges. (**b**) FTIR transmission spectra of Cu$_3$N. (**c**) Raman spectra of Cu$_3$N. (**d**) Wemple–DiDomenico plots.

Dielectrics and semiconductors are generally transparent at near-IR (NIR) wavelengths. These materials absorb light in the UV and visible ranges due to valence–electron transitions. Above the bandgap, we find interband transitions from the valence to the conduction bands, with the corresponding absorption of photons at energies higher than that of such a bandgap. Many will also show IR absorption due to the presence of intraband transitions within the valence band, molecular vibrations, phonons, or free charge carriers. In our Cu$_3$N samples, the free-carrier Drude model clearly failed to accurately predict the IR response. Therefore, the associated free charge carrier density must be necessarily smaller than

approximately $10^{17}$ cm$^{-3}$, the corresponding detection limit for the IR-VASE ellipsometric technique. Lastly, Figure 3c–f show the very-broadband complex refractive index, $\tilde{n}$, for Cu$_3$N from the UV to the MIR, reported for the first time, and clearly illustrates the aforementioned UV-Visible absorption due to the valence–electron transitions; moreover, copper-nitride is quasi-transparent across the remaining visible and NIR regions, until the presence of phonon *does* occur in the middle-infrared range of the spectra of the extinction coefficient, $k$, displayed in Figure 3c–f.

The crystal structure of Cu$_3$N belongs to the space group, Pm$\bar{3}$m, where the unit cell contains one formula unit. According to the space group theory, there are 12 phonon modes (lattice vibrations) at the center of the first Brillouin zone (Γ critical point), among which 9 are optic modes with irreducible representation: $\Gamma = 2T_{1u} + 1T_{2u}$. The $T_{1u}$ are infrared active, and the $T_{2u}$ modes are optically inactive (silent modes). Yu et al. [20] calculated the frequencies of Cu$_3$N Γ-point optical modes, and found that a peak at around 651 cm$^{-1}$ is associated with the Cu-N high-frequency stretching mode, $T_{1u}$, while another band at about 154 cm$^{-1}$ corresponds to the Cu-N-Cu bending mode, $T_{2u}$.

Concerning the previously mentioned UV-Visible absorption edge, Figure 4a displays the graph of the absorption coefficient spectrum, $\alpha(E)$, *versus* photon energy, for the three Cu$_3$N samples, calculated from both ellipsometric and intensity transmission measurements (WVASE multiple-sample analysis). These plots allow us to determine the *iso-absorption gap*, $E_{04}$, the energy value at which $\alpha = 10^4$ cm$^{-1}$. The obtained values of structural-defect-related Urbach energy parameter, $E_u$ [21–23], are also listed in Table 1. According to the literature, Cu$_3$N shows $E_u$ values ranging from 105 to 238 meV [5]; therefore, our obtained values of $E_u$ are closely within this reported range. The iso-absorption gap, $E_{04}$, for being empirical, is less sensitive to interpretational difficulties associated with the optical bandgap, and therefore, is in use as a common alternative and practical definition of the optical bandgap in poly-crystalline and non-crystalline semiconductors.

It should be emphasized that the values of the index of refraction reported in our present ellipsometric study are clearly much higher than those measured for Cu$_3$N with the prism coupling technique [24]. We considered that the values of the refractive index found with the latter technique, surprisingly, around 1.5 at four wavelengths in the NIR region, are notably underestimated, taking into account that the values of the refractive index determined in our study are very consistent with those previously reported in the literature [25], calculated making use of the popular Swanepoel transmission-envelope method.

### 3.3. FTIR and Raman Analysis

For an illustrative comparison, the Cu$_3$N thin-film samples were also analyzed by FTIR transmission spectroscopy. The representative FTIR-transmission spectra are shown in Figure 4b. The positions of the corresponding Cu$_3$N-phonon mode are all of them at around 645 cm$^{-1}$, in excellent agreement indeed with those independently calculated by infrared ellipsometry (i.e., this single band confirmed the creation of the Cu-N chemical bond). This would indicate that the amount of nitrogen contained within the sputtering-gas atmosphere was adequate to form the Cu$_3$N phase. A weak peak around 835 cm$^{-1}$, assigned to the Cu-N$_3$ chemical bond, was also observed in the two representative cases. In addition, a peak at 2049 cm$^{-1}$ also appeared in the FTIR-transmission spectra, corresponding to the stretching vibration of the N$_3$-azide.

Figure 4c, on the other hand, displays the representative Raman spectra of two of the Cu$_3$N thin layers. Notably, we have observed Raman shifts at 638 cm$^{-1}$ and 630 cm$^{-1}$, respectively, which are characteristic values of the Raman shift associated with Cu$_3$N thin film [5]. It has to be pointed out that, although the first-order Raman signal for a perfect crystalline Cu$_3$N is not allowed, Raman modes can be activated in the presence of structural disorder (e.g., a small crystalline size and/or the presence of structural defects). The increase found in Urbach energy, $E_u$, is primarily related to the increase in those structural defects (see Table 1).

### 3.4. Using the Sub-Gap Wemple–DiDomenico Single-Oscillator Dispersion Model

We focus next on fitting the obtained $Cu_3N$ refractive-index dispersion below the bandgap to the Wemple–DiDomenico single-effective-oscillator expression [26–29]:

$$n^2(E) - 1 = \frac{E_0 E_d}{E_0^2 - E^2}, \tag{1}$$

where $E_0$ is the energy of the *effective* dispersion oscillator, and $E_d$ is the dispersion energy or oscillator strength. By plotting $(n^2 - 1)^{-1}$ versus $E^2$ (Figure 4d), the two parameters $E_0$ and $E_d$ were determined. The obtained values of these Wemple–DiDomenico dispersion parameters, $E_0$ and $E_d$, are all indicated in Figure 4d. The oscillator energy $E_0$ is considered an 'average' energy gap. For the dispersion energy, $E_d$, on the other hand, a relationship was proposed [27]:

$$E_d(eV) = \beta N_c Z_a N_e, \tag{2}$$

where $\beta$ is a two-valued constant, $0.37 \pm 0.04$ eV for covalent materials, as in the case of the copper nitride under study, and $0.26 \pm 0.03$ eV for more ionic materials. $N_c$ is the coordination number of the cation nearest neighbor to the anion (copper in our case, with $N_c = 2$), $Z_a$ is the formal valency of the anion (nitrogen in our binary compound, with $Z_a = 3$), and $N_e$ is the *effective* number of valence electrons per nitrogen anion. In the $Cu_3N$ binary compound,

$$N_e^{Cu_3N} = \frac{(3 \text{ valence–electrons for 3 Cu-cations}) + (5 \text{ valence–electrons for 1 N-anion})}{(1 \text{ N-anion})} = 8.00. \tag{3}$$

We are not including the Cu *d*-electrons in our 'electron count' [27,30]. For this $Cu_3N$ material, we do *not* expect the $d^{10}$-core electron contribution, as observed in Cu halides: it would imply that we have to necessarily add 10 more electrons, for each of the copper cation atom's electron cloud to the present 'electron count', $N_e$. This would give rise to an enormous disagreement between the experimental and calculated values of the dispersion-energy parameter, $E_d$, as will be shown next.

The particular value of $E_d$ in copper nitride calculated by the use of Equation (3), is found to be 17.8 eV. The small differences with the experimental values of $E_d$ presented in Figure 4d, and especially in the most discrepant case of sample #1360, can reasonably be explained by the reported lack of stoichiometry of the studied sputtered $Cu_3N$ films (the Cu/N-ratio was found to be smaller than the expected ratio of 3) [4]. Moreover, the long-wavelength value of the refractive index, $n(E = 0)$, also displayed in Figure 3c–f, is given by the following expression:

$$n(0) = \sqrt{1 + \frac{E_d}{E_0}}. \tag{4}$$

Significantly, the values of these *static* refractive indices are all of them consistent with the data independently obtained by the IR-VASE ellipsometry (see the values of $n(E)$ in Figure 3c–f). In addition, it also seems reasonable to propose that the determined values of $n(0)$ increase with the mass density of the $Cu_3N$ samples. The less dense $Cu_3N$ film is the specimen #1360, with R = 0.8, according to its value of $n(0)$, 2.304, and the denser layer is the sample #1490, with R = 1.0, whose value of the static refractive index has been found to be 2.496. The 'in-between' case is the one of the film #1460, its value of R being 1.0, with $n(0) = 2.377$.

Furthermore, the existing correspondence between the Wemple–DiDomenico oscillator-energy parameter, $E_0$, and the so-called 'Wemple–DiDomenico gap', $E_g^{WD}$, is generally expressed as $E_0 \approx 2 \times E_g^{WD}$ [31,32]. For the $Cu_3N$ compound, the value of $E_g^{WD}$ obtained from the dispersion parameter $E_0$ goes from 1.50 eV for sample #1360 to 1.73 eV for sample #1460, very close to the indicated values of the iso-absorption gap, $E_{04}$.

### 4. Concluding Remarks

This investigation has unambiguously demonstrated the usefulness of the very-wide-spectral coverage (0.2–40 μm), of state-of-art spectroscopic ellipsometry, allowing the highly accurate determination of the complex refractive index, $\tilde{n} = n + ik$, in the whole UV-MIR spectral range, for the first time, using only just one technique. The novelty of the addition of the normal-incidence optical transmission has increased sensitivity to small UV-Visible absorption features in our reactive-RF-magnetron-sputtered $Cu_3N$ thin layers, deposited onto room-temperature glass and silicon substrates. The $Cu_3N$ samples investigated exhibit valence–electron transitions to energies above and below the bandgap that can all be successfully represented by a set of Gaussian oscillators. Based on FIB-SEM microscopy, the $Cu_3N$ micro-structural features were described in detail, following the well-known Thornton structural zone model. The investigated $Cu_3N$ layers were found to have a low porosity, which is certainly very convenient for PV applications.

Also, the alternate practical iso-absorption gap, $E_{04}$ (thus, avoiding the use of the sometimes ill-defined optical-bandgap parameter), exhibited a strong dependence upon growth conditions. It must be pointed out that the obtained values of $E_{04}$, interestingly, are found to be very close indeed to those of the previously reported indirect bandgap [5], thus indicating that they are mutually corroborated. Hence, we can conclude that a semiconductor material with values of $E_{04}$ between 1.3 and 1.8 eV could be considered suitable as a solar light absorber. Last but not least, it is worth mentioning that a standard optical gap of approximately 1.5 eV is regarded as an ideal and optimum value for the solar spectrum in a PV cell. Future efforts will be directed towards analyzing the photoresponse of our $Cu_3N$ material, building upon the contributions of this current work. We have presented a comprehensive study of the key UV-MIR optical properties of this material, previously unexplored in the literature, examining the effects of both types of gas used in its fabrication and its film thickness.

**Author Contributions:** E.M.: conceptulization, methodology, writing (original draft). E.B.: methodology, formal analysis, software. J.M.M.: methodology, formal analysis, software. M.B.: software, visualization, writing (review and editing). M.G.-G.: software, visualization, writing (review and editing). S.M.F.: investigation, data curation, resources, funding acquisition. M.I.R.-T.: investigation, data curation, funding acquisition. F.W.: supervision, validation, writting (review and editing). A.K.K.: supervision, validation, writing, (review and editing). All authors have read and agreed to the published version of the manuscript.

**Funding:** This study received financial support from MCIN/AEI/10.13039/501100011033, under grant PID2019-109215RB-C42. This funding is part of the economic recovery investment and reform measures under the Next Generation EU.

**Institutional Review Board Statement:** Not applicable.

**Informed Consent Statement:** Not applicable.

**Data Availability Statement:** The data employed in this study can be obtained from the corresponding author upon request.

**Acknowledgments:** The authors thank L. González-Souto for their invaluable assistance. J.M. Mánuel wishes to express gratitude to the "Central Service for Research in Science and Technology" (SC-ICYT) at the University of Cádiz.

**Conflicts of Interest:** The authors declare no conflict of interest.

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
