# Peer review of "Mid-Infrared (MIR) Complex Refractive Index Spectra of Polycrystalline Copper-Nitride Films by IR-VASE Ellipsometry and Their FIB-SEM Porosity"

_coatings, doi:10.3390/coatings14010005_

Round 1

Reviewer 1 Report

Comments and Suggestions for Authors

The authors fabricated Cu3N thin-film using the reactive sputtering technique and systematically characterize its material properties. It is of course important to explore the basic material properties of Cu3N if it is expected to be used in real applications. However, there are a few comments need to be properly addressed before this manuscript could be considered for publication.

1.     In the Abstract, there is no information about the characterization results. Please provide them, otherwise the readers only could know what you have done but without any results.

2.     In the Introduction, the importance of Cu3N semiconductor material has no been mentioned at all. What are the advantages of Cu3N in solar cell application?

3.     For the reactive sputtering, why are pure N2 gas and mixed N2+Ar chosen? Generally, Ar gas should be included as the carrying gas in sputtering.

4.     It is clear that the sputtered Cu3N thin-film is porous from the SEM images. Would this mean that, the sputtered Cu3N is non-reliable?

5.     Line 72. What is the meaning of Ts/Tm ratio? There is no definition of it.

6.     Though the authors systematically characterize the fabricated Cu3N material, could the authors provide some optoelectronic properties of Cu3N material, for example, the photoresponse of Cu3N/Si heterojunction? So the readers could see the potential of Cu3N in solar cell applications.

Comments on the Quality of English Language

Moderate editing of English language required

Reviewer 2 Report

Comments and Suggestions for Authors

Overall, this study aims to contribute to the development of environmentally friendly, advanced materials for sustainable energy. The optical properties of polycrystalline copper-nitride (Cu3N) thin films were studied. They have the potential to be used as low-cost absorber materials for photovoltaic cells. UV-MIR spectroscopic ellipsometry to determine the complex refractive index spectra of the films, which provides interesting information about the material's chemical and structural properties. The porous structure of the films was also investigated using focused ion beam scanning electron microscopy.  The films were found to have low porosity, which is desirable for photovoltaic applications. I believe that enough research has been done for a communication study. I think there are a few points that need to be corrected before the study is published.

-       When referring to figures, it is written as "fig" and "figure". Please be consistent.

-       Technical details about the radio frequency magnetron sputtering system should be added.  Dimensions, RF frequency information, etc. If possible, a schematic image can also be added.

-       On page 2, line 51, when referring to the imagej program, the spelling of the letter "j" should be corrected.

Reviewer 3 Report

Comments and Suggestions for Authors

The present paper is showing an interesting study of the optical properties obtained on the UV-MIR domain of the RF-magnetron-sputtered Cu3N thin layers grown at room temperature. The spectroscopic ellipsometry analysis on a very-wide-spectral range (0.2 – 40 μm) and the preparation of thin layers at room temperature are the strong points of this work, relevant for the field.

 I have a few questions:

1)       Did the authors perform XRD studies on this material that sustain the affirmation that this sample is of polycrystalline nature (as written in the title)?

2)       In the affirmation “Notably, we have observed Raman shifts at 638 cm1 and 630 cm1, respectively, which are characteristic values of the Raman shift associated with Cu3N.”, a reference could be added.

After answering to these demands, this work can be accepted for publication.

Author Response

Thank you for the comments. Please note we have addressed the two mentioned points:

1) We have added a new phrase (149-153) and reference [5] to support the affirmation of the poly-crystalline nature of the samples.

2) We have also added reference [5] to line 206, as indicated by the reviewer. 

Best regards,

The authors 

Round 2

Reviewer 1 Report

Comments and Suggestions for Authors

The authors have addressed all the comments and the manuscript is now acceptable for publication in its current form.

Author Response

Thank you for your comments, we will then go ahead and submit.